# Idiopathic Inflammatory Myopathies: Recent Evidence Linking Pathogenesis and Clinical Features

**DOI:** 10.3390/ijms26073302

**Published:** 2025-04-02

**Authors:** Eunice Fragoso Martins, Carla Helena Cappello, Samuel Katsuyuki Shinjo, Simone Appenzeller, Jean Marcos de Souza

**Affiliations:** 1Department of Internal Medicine, School of Medical Sciences, Universidade Estadual de Campinas (UNICAMP), Campinas 13083-881, Brazil; 2Post-Graduate Program in Medical Sciences, School of Medical Sciences, Universidade Estadual de Campinas (UNICAMP), Campinas 13083-887, Brazil; 3Department of Orthopedics, Rheumatology and Traumatology, School of Medical Sciences, Universidade Estadual de Campinas (UNICAMP), Campinas 13083-887, Brazil; 4Post-Graduate Program in Child and Adolescent Health, School of Medical Sciences, Universidade Estadual de Campinas (UNICAMP), Campinas 13083-888, Brazil; 5Division of Rheumatology, Faculdade de Medicina FMUSP, Universidade de São Paulo, São Paulo 01246-903, Brazil

**Keywords:** dermatomyositis, anti-synthetase syndrome, immune-mediated necrotizing myopathy, inclusion body myositis, pathogenesis

## Abstract

Idiopathic inflammatory myopathies are rare and complex representatives of systemic connective tissue diseases. Described initially as only two entities, recent advances in molecular and imaging techniques now divide them into many subtypes, each with unique pathogenesis and clinical phenotypes. Dermatomyositis and its juvenile form are the most prevalent subtypes and are characterized by systemic vasculopathy and humoral autoimmunity. Genetic predisposition and environmental triggers initiate immune tolerance breakdown, leading to autoantibody production, complement activation, and tissue damage. Anti-synthetase syndrome primarily affects the lungs, where immune responses to aminoacyl-RNA synthetases drive vasculopathy, lung inflammation, and fibrosis. Immune-mediated necrotizing myopathies are muscle-specific, with autoantibodies inducing fiber necrosis and atrophy. Lastly, sporadic inclusion body myositis is a slowly progressive myopathy in which dysfunctional protein handling and autophagy are more important pathogenic elements than muscle inflammation itself. The expanding body of basic science evidence can be overwhelming, making it challenging to connect pathogenic mechanisms to clinical manifestations. This review aims to address this challenge by presenting recent insights into myositis pathogenesis from a practical perspective, reinforcing the links between basic science and clinical semiology.

## 1. Introduction

Among systemic autoimmune rheumatic diseases, idiopathic inflammatory myopathies (IIM), or systemic autoimmune myopathies, represent a family—or even subfamilies—of heterogeneous diseases that share striated skeletal muscle inflammation as a common trait. Muscle weakness is the classical clinical manifestation of IIM; however, other organs and systems can also be affected, particularly the skin, lungs, and blood vessels [1,2].

Recent advances in molecular features and novel autoantibodies have provided significant insights into the pathogenesis of IIM, offering clues for future treatment strategies. However, descriptions correlating pathogenesis with clinical manifestations remain relatively rare in the scientific literature. For healthcare providers, even a superficial understanding of the immunological and inflammatory mechanisms underlying phenotypic presentations can offer valuable guidance for selecting the most appropriate interventions for each patient. This is especially important in a disease group with dozens of reported antibodies [1] and heterogeneous correlations owing to their low prevalence.

In this review, we aimed to highlight recent evidence on the pathogenesis of dermatomyositis (DM), juvenile idiopathic inflammatory myopathy (JIIM), anti-synthetase syndrome (ASyS), immune-mediated necrotizing myopathy (IMNM), and sporadic inclusion body myositis (sIBM), while also correlating these findings with clinical manifestations.

### 1.1. Dermatomyositis (DM)

Among IIMs, DM is the most common subtype, accounting for approximately 30–40% of these cases [3,4,5]. DM is clinically characterized by skin involvement; other extramuscular features, such as interstitial lung disease (ILD); proximal muscle weakness, predominantly in combination with a typical skin rash, particularly Gottron’s papules/signs or heliotrope erythema, which leads to difficulties in swallowing, running, climbing stairs, standing up from the floor, and/or raising the arms [3]; elevated serum creatine kinase (CK) activity, which is frequently increased; and myositis-specific autoantibodies (MSAs) (anti-MDA-5, anti-NXP-2, anti-Mi-2, anti-TIF-1γ, and anti-SAE), which can be found in approximately 60% of patients [1,6]. Moreover, DM is associated with the presence or development of cancer in approximately 8% of patients (standardized incidence ratio 2.4–6.2) [7,8].

The cause of DM is attributed to a combination of environmental (e.g., viruses such as human T-lymphotropic virus type 1 [HTLV-1] and ultraviolet sunlight exposure) and genetic factors (e.g., human leukocyte antigen-HLA class II alleles, driven by T cells). There are differences in genetic associations, with greater susceptibility in White populations associated with HLA DRB1*0301 and DQA1*0501 [9] and an association with HLA-B7 in Asian patients [10]. HLA DRB1*0301 or DQA1*0501, when coupled with the anti-Jo-1 autoantibody, and DRB1*07 or DQA1*0201 paired with anti-Mi-2 antibodies, have even stronger associations [10].

The current pathogenesis of DM is attributed to immune and non-immune mediators. Muscle damage in DM is believed to be mediated by humoral factors (antibodies and the complement system) directed against endomysial capillary endothelial cells. Given the presence of T cells in DM muscle biopsies and HLA-DR association, it can be hypothesized that both adaptive and innate immunity play a role in the development of muscle inflammation, as DM inflammatory infiltrates are often composed mainly of CD4+ T cells and macrophages with a perivascular location, particularly in the perimysium [10,11,12]. The disease begins with the presence of microtubular inclusions in endothelial cells, often preceding inflammatory cell infiltration [13,14]. These inclusions are related to the endoplasmic reticulum (ER) or outer nuclear membrane and likely represent membrane specializations within the ER during certain stages of cellular activity [15]. Deposition of the membrane attack complex (MAC) on the walls of endomysial capillaries has high sensitivity and specificity for DM diagnosis, distinguishing it from other IIM subtypes [16].

The role of mitochondrial dysfunction in the pathogenesis of DM has been a growing area of interest [17]. This assumption is supported by (1) mitochondrial DNA (mtDNA) gene variants [18] and mtDNA depletion [19] reported in DM patients, particularly in the perifascicular region, where vasculopathy is most prominent; (2) the presence of surrogate markers of oxidative phosphorylation impairment, such as reduced ATP production and proton efflux from muscle fibers [20]; (3) decreased expression of genes related to electron transport chain complexes in DM [21], along with an increased proportion of cytochrome C oxidase (COX)-negative fibers, the latter correlating with reduced aerobic capacity [21]; (4) in animal models of DM, a correlation between the IFNγ cytokine profile—one of the inflammatory signatures in DM [22]—and reduced mitochondrial gene expression, as well as diminished expression of genes related to oxidative phosphorylation following IFNγ exposure [23]; and (5) the presence of autoantibodies directed against mitochondria in IIM [24,25,26], although not always associated with mitochondrial dysfunction [27].

Despite the compelling evidence linking mitochondrial dysfunction to DM, it remains arguable whether the immune attack is the primary driver of mitochondrial homeostasis disruption, leading to reactive oxygen species-mediated injury, dysfunction, and even organelle death. When mtDNA and intracellular components are presented to pattern recognition receptors, type I IFN may exacerbate inflammation [17] and potentially contribute to the development of autoantibodies.

The antibodies present in DM include anti-TIF-1γ, discovered in 2006 by Targoff et al. as a newly recognized autoantibody targeting a 155/140-kDa protein, which is particularly associated with cancer in DM [28]. The expression of the Mi-2 autoantigen is significantly increased (approximately tenfold) in muscle biopsies from patients with DM only [10]. This demonstrates that a specific pattern of autoantigen expression in DM correlates with a specific autoantibody response in the disease. These antigenic autoantigens elicit a pro-inflammatory response, subsequently attracting immune and inflammatory cells [10]. The cytokine involved in DM is the pro-inflammatory molecule high-mobility group box 1 (HMGB1). Although it is an omnipresent nuclear DNA-binding protein, it can also be released by any damaged cell or activated macrophage. Therefore, HMGB1 released from injured cells can induce and perpetuate inflammation [29].

Regarding the treatment of DM, chronic immunosuppressive therapy is necessary due to continuous disease activity [30,31]. The current first-line therapy consists of glucocorticoids, such as high-dose prednisone (1–1.5 mg/kg/day), methylprednisolone, or dexamethasone, for all IIM subtypes, including DM [31,32]. However, no pharmacological treatment for IIMs can be recommended based on randomized clinical trials, except for intravenous human immunoglobulin (IVIg), which is used as adjunct therapy for refractory DM [33].

Figure 1 illustrates the main elements involved in DM pathogenesis and their clinical associations.

### 1.2. Juvenile Idiopathic Inflammatory Myopathies (JIIM)

JIIM are a group of rare systemic autoimmune diseases that have in common the presence of vasculopathy and endothelial dysfunction [34,40,41]. Clinically, muscle and skin involvement are hallmarks of the disease; however, other internal organs, such as the lungs, joints, and gut, can also be affected [34,40,41].

Based on clinical and histopathological findings, JIIM can be divided into subtypes. Juvenile dermatomyositis (JDM) accounts for approximately 80% of JIIM cases, followed by ASyS, IMNM, overlap syndromes, and amyopathic myositis [42,43].

The pathogenesis of JIIM involves a complex interplay between genetic and epigenetic factors, leading to innate and adaptive immunological activation, in addition to vascular and metabolic dysfunction, which are responsible for diverse clinical presentations [34].

In both adult and pediatric IIM, the strongest genetic association in White populations is within the AH8.1 (HLA A1-B8-DR3-DQ2) haplotype of the major histocompatibility complex (MHC) [44,45]. In contrast to adult-onset disease, the allele HLA-DRB1*0301 and amino acid position 37 within HLA-DRB1 confer risk factors for JIIM [46].

Genetic specificity has also been observed for anti-3-hydroxy-3-methyl-glutaryl-coenzyme A reductase (HMGR) antibodies. In childhood-onset disease, positive anti-HMGR is associated with HLA-DRB1*0701, whereas adult-onset disease is associated with HLA-DRB1*1101 [47]. Independent of AH8.1, other non-MHC genetic loci, including variants in PTPN22 and rs2304256 in TYK2, confer additional risk to both adults and JIIM [48,49].

Considering the epigenetic factors associated with JIIM, ultraviolet light intensity and exposure have been associated with both disease onset and severity [50,51,52]. Infectious agents, such as streptococcal infections, picornavirus, enterovirus, and mycoplasma, have been implicated in its pathogenesis, though with controversial results [53,54,55].

The type 1 interferon (IFN) signature is a key feature of JIIM and is associated with pathogenic changes in several diseases. In muscle tissue, type 1 IFN leads to the overexpression of MHC proteins and ER stress. As a consequence, an inflammatory cascade is initiated and perpetuated through the nuclear factor kappa B (NF-κB) pathway [34]. Type 1 IFN signaling also leads to neutrophil extracellular trap (NET) formation and mitochondrial dysfunction [39]. Increased NET formation is associated with calcinosis in JIIM [39].

B and T cells are activated and associated with the clinical course of the disease. Autoreactive B cells produce MSAs that are associated with distinct clinical and laboratory features (Table 1) [35]. Regulatory B (Breg) cells have a pro-inflammatory phenotype characterized by increased production of IL-6, galectin-9, and CXCL10 [56,57]. Studies have shown that these inflammatory markers are correlated with disease activity [34,57].

T cell dysfunction includes a shift in the T cell compartment toward a T helper 2 (Th2) and T helper 17 (Th17) cell phenotype, increasing MSA production [58,59]. An increased number of circulating endothelial cells is associated with disease activity, muscle capillary loss, and nailfold capillary abnormalities [37].

Activated endothelial cells secrete soluble endothelial adhesion molecules, increasing inflammation and complement deposition on capillaries, which contributes to the vasculopathy observed in JIIM [36,37].

As DM and JDM share many pathogenic aspects, treatment follows the same principles. A combination of oral corticosteroids and methotrexate is a reasonable first-line treatment for most cases [60]. IVIg is effective for rescue induction, as are methylprednisolone pulse therapy and rituximab [60].

**Table 1 ijms-26-03302-t001:** Differences in the clinical associations of myositis-specific and myositis-associated autoantibodies between juvenile idiopathic inflammatory myopathies and their adult counterparts [34,35,51,61,62,63,64,65].

Autoantibody	Frequency in JIIM	Importance in JIIM	Clinical and Laboratory Features in JIIM	Frequency in Adult Myositis	Importance in Adult Myositis	Clinical and Laboratory Features in Adult
Anti-TIF-1γ	17–35%	↑ White, younger patients, polycyclic disease, and no association with malignancy	More severe cutaneous disease (cutaneous ulceration, photosensitivity, and lipodystrophy)	10–20%	Associated with malignancy	Classical dermatologic manifestations, often clinically amyopathic
Anti-NXP-2	15–25%	↑ White, younger patients, chronic course, and no association with malignancy	Calcinosis, prominent muscle weakness, joint contractures, dysphagia, and dysphonia.	3–20%	Associated with malignancy	Extensive skin disease with calcinosis
Anti-MDA-5	6–38%	Older Japanese children	Mild muscle disease, constitutional symptoms ↑ risk of cutaneous and oral ulceration, arthritis and lung disease	10–30%	Possibly aggressive and highly morbid	Clinically amyopathic with skin ulcers and rapidly progressive interstitial lung disease
Anti-Mi-2	4–10%	Older Hispanic children	Pharyngeal weakness or dysphagia, edema, and cutaneous features, with a low risk of lung disease and a relatively favourable prognosis	5–20%	Relatively favourable prognosis	Classical skin findings with muscle weakness
Anti-SAE	0.3–9.1%	Predominant cutaneous involvement	Predominant cutaneous involvement	<10%	Associated with malignancy	Extensive skin disease with late-onset myositis
Anti-synthetase	2–5%	Older children of African descent	Higher risk of lung disease and mortality	10–30%	Not associated with malignancy	Almost universal lung disease. Variable muscle and skin disease
Anti-SRP	1.6–4%	Older children of African descent	Higher risk of dysphagia, joint contractures, and cardiac involvement	5–15%	Possible myocardial involvement	Prominent muscle weakness and atrophy
Anti-HMGCR	Very rare	Chronic disease	Severe proximal muscle weakness, joint contractures, dysphagia, and very high serum creatinine kinase levels	6–10%	Previous use of statins	Often restricted to skeletal muscle. Weakness of pelvic and scapular girdles

### 1.3. Anti-Synthetase Syndrome (ASyS)

ASyS is characterized by a triad of clinical manifestations comprising myositis, arthritis, and predominantly ILD, in addition to fever, Raynaud’s phenomenon, and “mechanics’ hands” [66]. From a laboratory perspective, ASyS is defined by the presence of autoantibodies targeting aminoacyl-tRNA synthetases (anti-ARS) [66,67]. To date, 10 distinct anti-ARS autoantibodies have been identified, with anti-histidyl (anti-Jo-1) being the most prevalent [68,69], followed by anti-threonyl (anti-PL-7) [69], anti-alanyl (anti-PL-12) [70], anti-glycyl (anti-EJ) [69], anti-isoleucyl (anti-OJ) [71], anti-asparaginyl (anti-KS) [72], anti-phenylalanyl (anti-Zo) [73], anti-tyrosyl (anti-Ha) [74], anti-valyl [75,76], and anti-cysteinyl [76]. Each of these anti-ARS autoantibodies may represent a distinct subcategory with a unique clinical presentation and may serve as a prognostic indicator for clinical complications and outcomes.

The pathogenic mechanisms of anti-ARS autoantibodies have not yet been fully elucidated. Given the high frequency of ILD, it has been hypothesized that the lungs may serve as the initiating site of the disease. Lung exposure to various agents, such as infections or tobacco [77], may act as a trigger in genetically susceptible individuals. In such cases, ARS and potential neoepitopes could become exposed to the immune system, leading to its activation, particularly in high-risk individuals, and initiating the adaptive immune response to produce anti-ARS autoantibodies.

Notably, histidyl-tRNA synthetase is highly expressed in the lungs and regenerating muscle fibers [29]. In the lungs, it can be cleaved by granzyme B, potentially revealing neoepitopes [78], and can act as a chemokine that attracts CCR5+ cells [79]. The presence of anti-Jo-1 autoantibodies in bronchoalveolar lavage fluid suggests local autoantibody production in the lungs, with some antibodies targeting the WHEP domain of histidyl-tRNA synthetase [80]. Additionally, the presence of germinal center-like structures in the lungs further supports this hypothesis [81].

ASyS is also characterized by an inflammatory response mediated by type II IFN-γ. IFN-γ is overexpressed in ASyS lungs and promotes elevated levels of IFN-γ-inducible chemokines, such as CXCL9 and CXCL10 [82]. These chemokines recruit activated T cells to the lungs [22,82,83]. Patients with ASyS and diffuse alveolar damage have higher serum levels of CXCL9 and CXCL10, emphasizing the role of these chemokines in pulmonary manifestations [22].

Regarding the pathophysiological aspects of muscle tissue, immune cell infiltration, including macrophages, T cells (CD4 and CD8), B cells, and plasma cells, occurs in the perimysium or endomysium [84,85]. These cells frequently cluster, suggesting cell-to-cell interactions [84,85]. This observation is further supported by the expression of chemoattractants on various immune cells that facilitate the homing of B cells and plasma cell subtypes [84,85].

A recent study demonstrated that IgG from Jo-1 autoantibody-positive patients can induce complement-dependent cellular cytotoxicity in human muscle microvascular endothelial cells [86]. The binding of Jo-1 antibodies to Jo-1 embedded in the muscle capillary cell membrane can lead to complement deposition, resulting in the lysis of endomysial capillaries and muscle ischemia [86]. This process contributes to perifascicular necrosis, which is characteristic of myositis in ASyS [86]. Unlike dermatomyositis, where complement primarily affects the capillary endothelium, in ASyS, complement is deposited in the sarcolemma [86]. These antibodies also induce the upregulation of triggering receptors expressed on myeloid cells (TREM-1) in perimysial blood vessels [87]. TREM-1 appears to initiate inflammation and promote the migration of inflammatory cells by inducing the secretion of proinflammatory cytokines such as tumor necrosis factor-alpha (TNFα) and chemokines [87,88].

Muscular features are also characterized by an inflammatory response mediated by type II IFN-γ, as evidenced by the increased expression of HLA-DR in the perifascicular region of muscle biopsies [22,89,90]. Unlike DM, ASyS does not typically exhibit sarcoplasmic expression of the MxA protein, which is a hallmark of type I IFN activity, suggesting a minor role for type I IFN in the muscular manifestations of ASyS [22,89,90].

Interleukin-17A (IL-17A), a proinflammatory cytokine produced by Th17 cells, appears to play a significant role in ASyS inflammation [91,92]. IL-17A stimulates the production of other proinflammatory cytokines, such as TNF-α, IL-1, and IL-6, which recruit inflammatory cells [92]. Behrens Pinto and colleagues examined the IL-17A serum levels in ASyS patients and found them to be elevated compared to healthy controls, although the levels did not correlate with disease activity [91].

Figure 2 illustrates the correlation between clinical manifestations and pathogenic features of ASyS.

### 1.4. Immune-Mediated Necrotizing Myopathy (IMNM)

IMNM is usually described, when compared to its other muscle-predominant counterparts among IIM, as a disease with more rapid-onset weakness, higher values of serum creatine phosphokinase, and higher chances of atrophy and fat deposition from the onset [94,95,96]. However, extramuscular manifestations are scarce or mild, raising the possibility of muscle-driven pathogenesis [97]. Currently, based on serological and prognostic factors, IMNM is subdivided according to the presence of anti-signal recognition particles (anti-SRP), anti-HMGR, or absence of autoantibodies (e.g., seronegative IMNM) [98]. Pathogenic mechanisms remain unclear for seronegative IMNM; therefore, our review focuses mainly on seropositive data.

Broadly, two important mechanisms have been consistently described in the pathogenesis of IMNM: (a) muscle fiber necrosis mediated by complement activation [98,99] and (b) dysfunctional muscle repair with early-onset muscle damage [100]. It is also noteworthy that treatment strategies aimed at controlling humoral mechanisms of autoimmunity, such as intravenous immunoglobulin and rituximab, represent important assets to control the disease [1,101], suggesting that antibodies are related to its pathogenesis [97].

In 2018, Allenbach et al. analyzed muscle samples from 44 patients and demonstrated that muscle fiber necrosis and regeneration were the main pathological findings, with infiltrates comprising mainly CD68+ macrophages and, to a lesser degree, CD3+ lymphocytes [98]. In addition, in the vicinity of necrotic fibers, immunohistochemistry and immunofluorescence for the membrane attack complex (C5b-9) and C1q, respectively, were significantly more positive than those in the controls [98]. Furthermore, Bergua et al. also demonstrated that transferring antibodies from anti-SRP and anti-HMGR patients to mice led to muscle impairment, which was less pronounced in C3-deficient mice and more pronounced with complement supplementation [102]. Taken together, these studies provide a solid argument involving antibody-mediated complement-induced necrosis in the pathogenesis of IMNM.

Despite the well-described role of complement in muscle damage in IMNM patients, it is remarkable that the constant activation of the complement system and massive tissue necrosis do not substantially recruit lymphocytes to the muscle, as is the case in DM and ASyS [34,84,85]. One possible explanation comes from a recent study reporting overexpression of programmed cell death protein 1 (PD-1) in muscle lymphocytes [103], leading to lymphocytic exhaustion. The main argument is that because the target of the disease (e.g., muscle constituents) is so abundant in the environment, lymphocytes may be directed towards apoptosis and anergy.

The effect of autoantibodies on muscle repair was also studied by Arouche-Delaperche et al. in 2017. They incubated mature human muscle cells with purified autoantibodies from IMNM-seropositive patients and found that exposure to the antibodies induced myotubule hypotrophy and augmented the expression of atrogenes MAFbx/ATROGIN-1 and TRIM63/MURF1 [100]. Among other functions, these genes contribute to muscle regeneration and are implicated in various conditions of muscle atrophy [104,105]. In addition, they also demonstrated that the antibodies decreased the expression of IL-4 and IL-13 in myoblasts, suggesting a role for cytokines in myotubule hypotrophy. These findings, albeit in vitro, might help to understand why patients with IMNM are so prone to muscle damage compared to other patients with IIM.

As autoantibodies are intrinsically related to IMNM pathogenesis, efforts have been made to understand the roles of antigens and the mechanism of immunotolerance loss. SRP is a ribonucleoprotein distributed in the cytoplasm of many cell types and is responsible for aiding the transportation of nascent polypeptides to the ER [106]. HMGR, on the other hand, is a glycoprotein located in the ER and is widely known for its function in cholesterol synthesis and for being a target of statins [107]. To account for the membrane expression of otherwise cytoplasmic antigens, the authors observed that regenerating fibers are able to express ER proteins on their surface and that immunohistochemistry with IMNM autoantibodies stains the sarcolemma [99,100]. Thus, it is most likely that an initial trigger is required, generating muscle damage and repair for those antigens to be expressed in the cell membrane. As a second step, genetically predisposed lymphocytes recognize these epitopes and generate a break in immunotolerance [99]. In the case of statin-induced myositis, this mechanism is particularly appealing because the drug is able to upregulate the HMGR levels [108].

As mentioned in the beginning of this section, prognostic factors divide anti-SRP and anti-HMGR patients. As pathogenesis is consistently related to the direct action of antibodies, researchers have aimed to understand the specific features of each. In 2022, Lia et al. described the pathological findings from a sample of patients with anti-HMGR-related IMNM, reporting a significant overexpression of VEGF-A and CXCL12 in macrophages dwelling in the perivascular region of muscle specimens [109]. Because these substances are involved in angiogenesis, the authors also found that the expression of these markers correlated well with the degree of neovascularization. Although preliminary, these data might help to comprehend why anti-HMGR patients present less long-term muscle damage than their anti-SRP counterparts.

Most patients with IMNM present with a relatively rapid course of proximal muscle weakness and, occasionally, myalgia and dysphagia [99]. Creatine kinase levels are usually very high [94], which is attributable to the necrotic features observed in IMNM [98]. Clusters of slowly progressive cases, particularly in younger individuals, have also been described [110]. Extramuscular disease is uncommon or mild, with interstitial lung disease being the most common, occurring in 16% of patients [94,111]. Myocardial involvement is described as part of the muscular spectrum, although it infrequently manifests as a clinically significant cardiac syndrome [111]. Subclinical myocardial involvement may be more common than previously thought, with up to 52% of patients exhibiting functional or electrical abnormalities in a recent report [112]. However, more studies on this subject are needed, as magnetic resonance findings in different populations did not corroborate these results [113,114]. The relative specificity of muscular manifestations is linked to the pathogenesis described earlier. It is thought that the initial trigger, occurring at the muscular level, is necessary to expose muscular antigens on the myocyte surface, allowing autoantibodies to form and act [99].

Disease severity, resistance to treatment, and degree of atrophy are more closely related to anti-SRP than anti-HMGR antibodies [94,115]. One possible explanation is provided earlier, referring to the favorable neovascularization phenomenon in the latter [109].

Most patients with IMNM will receive glucocorticoids, but the majority will require a second drug to induce remission and reestablish muscle strength [75]. Given the need to control complement-mediated cell death, IVIg is a reasonable approach for refractory patients and should be considered early in the disease course [77].

Figure 3 synthesizes the correlation between clinical manifestations and pathogenic features of IMNM.

### 1.5. Sporadic Inclusion Body Myositis (sIBM)

IBM represents a broad family of myopathies characterized by rimmed sarcoplasmic vacuoles as a histological feature [117]. Many of these myopathies are not immune-mediated and have a known pattern of inheritance, such as GNE or VCP mutations [117]. sIBM, on the other hand, is a single immune-mediated disease classified as a member of the IIM family. Patients with sIBM are typically middle-aged and present with slow-onset muscle weakness and atrophy of the quadriceps and finger flexors [118,119]. Mechanisms involved in the pathogenesis of sIBM include not only T cell-driven inflammation [117,120] but also various degenerative processes, such as misfolded proteins, autophagy impairment, oxidative stress, and mitochondrial dysfunction [121].

It is well known that muscle fibers in sIBM are predominantly invaded by cytotoxic CD8+ lymphocytes [122,123]. Macrophages, dendritic cells, and CD4+ T lymphocytes are also present but at lower frequencies [117,124]. Lymphocytes in sIBM expand clonally in the muscle and express unique Vβ genes [125,126], suggesting that muscle proteins are locally presented from MHC class I to local lymphocytes. Therefore, immunotolerance appears to be broken locally, but no specific antigen target has been consistently identified [117,124], although the serological biomarker of sIBM, cytosolic 5′-nucleotidase A1 (cNA1) [127,128], is a potential culprit. Continuous antigen presentation in the muscle also promotes T cell exhaustion, as suggested by a large proportion of T cell phenotypes associated with reduced proliferation abilities [129], as well as increased expression of PD-1 [103]. This apparent regulatory feature of sIBM suggests a disease in which inflammation is not out of control but, rather, degeneration predominates, which might help explain why immunosuppression is rarely beneficial.

A humoral response has been described in sIBM, as CD138+ plasma cells are increased in the muscle of these patients [130], and B-cell activation factor (BAFF) levels are elevated in their serum [131]. However, the most convincing evidence comes from the discovery of a current antibody related to sIBM, cNA1 [128], which is present in approximately 30% of patients with sIBM, with a low prevalence in other myopathies [132]. This enzyme is highly expressed in skeletal muscle and is responsible for the dephosphorylation of adenosine monophosphate into adenosine and phosphate [128,132].

Protein misfolding is a well-described phenomenon in sIBM, as aggregates of various proteins, including phosphorylated tau and beta-amyloid, have been consistently observed in the muscle and serum of patients [133,134,135,136]. Dysfunction of autophagy, the ubiquitin-proteasome system (UPS), and oxidative stress likely play a role in this intricate chain of events. Accumulation of proteasome subunits and decreased proteasomal proteolytic activity have already been described in the muscle of subjects with sIBM [137], corroborating UPS impairment. Increasing evidence also suggests that lysosomal activity is elevated in sIBM [138,139], coupled with decreased lysosomal enzymatic activity [140]. This suggests not only overproduction of protein aggregates but also a failed disposal mechanism [124].

Mitochondrial disease is also a common feature of sIBM [141]. Histologically, muscle tissue from patients shows accumulation of defective mitochondria in the form of subsarcolemmal deposits seen in specific colorations [141,142]. In addition, cytochrome C oxidase (COX)-negative fibers are also found in some muscle samples, suggesting an impairment in energy production [124,141]. Importantly, malfunctioning of oxidative complexes within the respiratory chain can generate reactive oxygen species (ROS), which are also responsible for enhancing oxidative stress and protein misfolding. Mechanisms leading to mitochondrial dysfunction are speculative but likely involve mitochondrial DNA deletions [143], reported in up to 60% of affected individuals [144], and aberrant mitochondrial microstructure [145]. Of note, as in DM, several danger-associated molecular patterns, including their highly methylated DNA, cardiolipin, and *N*-formyl methionine peptides, can initiate or enhance the type I IFN response, perpetuating the inflammatory cycle [17].

Clinically, sIBM has a weakness pattern that is distinct from that of its IIM counterparts. While gluteal, hip flexors, and shoulder girdle muscles are commonly the most affected muscles in IIM in general, sIBM preferentially affects the quadriceps (with late rectus femoris weakness), fibularis, gastrocnemius, biceps brachialis, and deep finger flexor muscles [117,146,147,148].

Although it may be challenging to isolate what is primarily related to the disease and what is secondary to inflammation and/or aging in sIBM, the contemporary concept is that sIBM is a pauci-inflammatory disease with multifactorial degenerative features. sIBM pathogenesis does not provide specific clues to its phenotypic presentation, except that extramuscular disease is uncommon, arguably because loss of immunotolerance occurs in situ (as mentioned previously).

Given its degenerative features, treating sIBM is challenging, and most pharmacological approaches have failed to produce significant gains in strength [149]. The current cornerstone of treatment is rehabilitation, aimed at least at slowing the rate of decline [149].

## 2. Conclusions

In summary, IIMs are rare pleomorphic diseases with multiple organ involvement. The complement pathway and its associated components play important roles in the pathogenesis of the most prevalent forms of IIM. The regenerative properties of the muscle are also impaired in sIBM and IMNM. Future therapeutic strategies should target humoral mechanisms of autoimmunity, as well as myoblast maturation and muscle autophagy pathways.

## Figures and Tables

**Figure 1 ijms-26-03302-f001:**
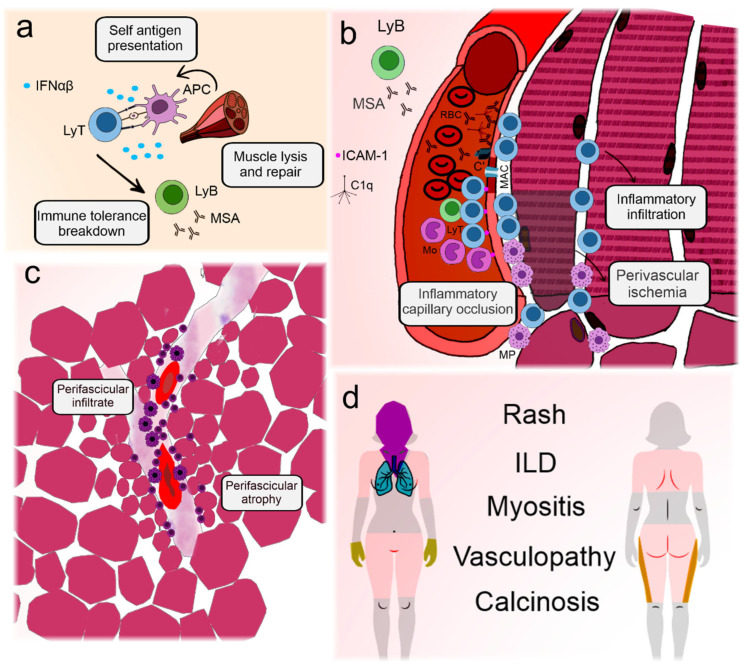
Illustration depicting the pathogenesis and key clinical features of dermatomyositis. Dermatomyositis and its juvenile form are the most prevalent idiopathic inflammatory myopathies. Although differences exist between adult and childhood-onset disease, they are archetypes of systemic vasculopathy with humoral autoimmunity [10]. (**a**) Presumable triggering agents, such as viruses or sunlight, promote antigen expression in genetically predisposed individuals with a type I interferon signature [34]. This leads to immune tolerance breakdown and production of myositis antibodies [35]. (**b**) Circulating autoantibodies are deposited on endothelial cells in the muscle and skin, initiating the complement cascade and formation of the membrane attack complex in the capillaries [16]. Aggression to the capillaries leads to the expression of adhesion molecules and recruitment of lymphocytes and macrophages that migrate to the perifascicular region [36,37]. Eventually, the capillaries may be completely occluded, leading to perifascicular ischemia [10]. (**c**) Because the blood vessel is the core of immune aggression, infiltrates tend to concentrate around them. Continuous ischemia impairs muscle regeneration and leads to perifascicular atrophy [10]. (**d**) The main clinical manifestations of dermatomyositis include polymorphic forms of rashes, representing perivascular dermatitis [10]. In the extremities, vasculopathy can lead to nailfold disease and skin ulceration [37]. Muscle manifestations are frequent and involve weakness, especially in the scapular and pelvic girdles and neck flexors. Lung disease is less well understood but may be related to the activation of lung macrophages and fibroblasts induced by neutrophil extracellular traps, which are activated by autoantibodies [38]. The latter is also associated with calcinosis [39]. **C’**: aggregated complement fractions; **C1q**: complement 1q component; **RBC**: red blood cell; **ICAM-1**: intercellular adhesion molecule 1; **IFNαβ**: Type I interferons alpha and beta; **ILD**: interstitial lung disease; **LyB**: B-lymphocyte; **MSA**: myositis-specific antibodies; **LyT**: T-lymphocyte; **APC**: antigen presenting cell; **MAC**: membrane attack complex; **Mo**: monocyte; **MP**: macrophage.

**Figure 2 ijms-26-03302-f002:**
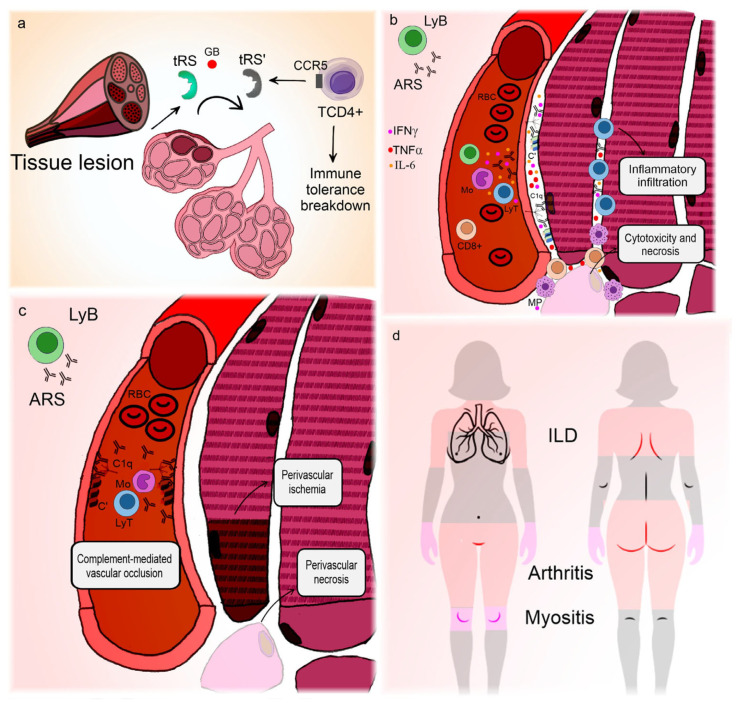
Pathogenesis and clinical manifestations of anti-synthetase syndrome. The lung is the main organ involved in anti-synthetase syndrome. (**a**) Environmental triggers cause lung injury [77] and expose wild aminoacyl-RNA-synthetases to granzyme B action, promoting the formation of neoantigens [78] that can act as chemokines for T-lymphocytes [79]. In genetically predisposed individuals, this can lead to immune tolerance breakdown and the production of anti-RNA-synthetases. (**b**) Anti-RNA-synthetases deposit in the sarcolemma [86] and, mediated by interferon γ [82], interleukin-6, tumor necrosis factor α, and possibly interleukin-17 [91,92], promote inflammatory infiltration, cytotoxicity, and myofiber necrosis. (**c**) Alternatively, anti-RNA-synthetases can deposit in blood vessels, causing complement-mediated vasculopathy [86]. (**d**) Interstitial lung disease is almost universal and represents the main site of neoantigen formation, so the lungs may harbor germinal center-like formations [81]. Vasculopathy is responsible not only for muscle ischemic dysfunction and perifascicular necrosis but also for the Raynaud phenomenon, which occurs in a large percentage of patients. Arthritis is frequent and can be erosive, though its pathogenesis is unclear, presumably related to humoral mechanisms. In contrast, thickening of the palms and soles is a clinical manifestation of skin hyperkeratosis, focal parakeratosis, and psoriasiform acanthosis [93]. Since mononuclear cell infiltrates around blood vessels usually accompany these alterations [93], one could hypothesize that they represent a dysfunctional form of skin regeneration. **ARS**: anti-RNA-synthetase antibodies; **C’**: aggregated complement fractions; **C1q**: complement 1q component; **CCR5**: C-C chemokine receptor type 5; **GB**: granzyme B; **IFNγ**: interferon gamma; **ILD**: interstitial lung disease; **LyB**: B-lymphocyte; **LyT**: T-lymphocyte; **Mo**: monocyte; **MP**: macrophage; **RBC**: red blood cell; **TCD4+**: CD4+ T-lymphocyte; **TNFα**: tumor necrosis factor alpha; **tRS**: amino-acyl-tRNA-synthetase; **tRS’**: modified neoantigen of amino-acyl-tRNA-synthetase.

**Figure 3 ijms-26-03302-f003:**
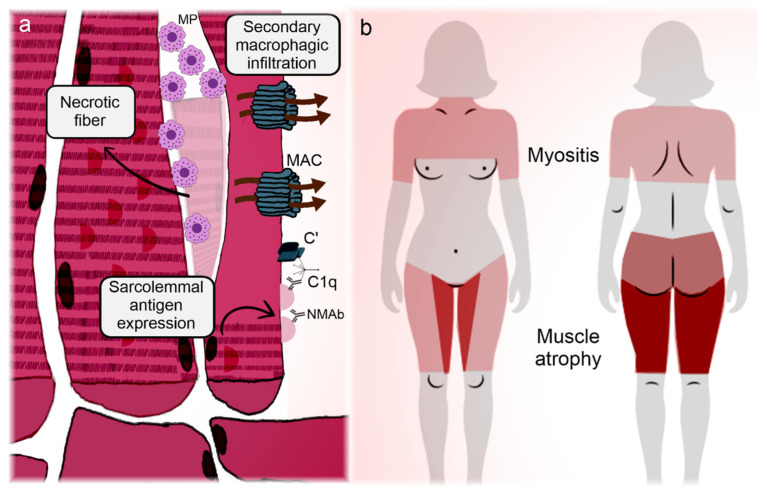
Illustration representing the main manifestations and pathogeneses of immune-mediated necrotizing myopathies. Immune-mediated necrotizing myopathies (IMNMs) are conditions relatively restricted to the muscle (skeletal and myocardial). (**a**) Regenerating fibers express the targets of the disease on their membrane, enabling the immune tolerance breakdown and subsequently deposition of autoantibodies on the sarcolemma [99,100]. In the case of 3-hydroxy-3-methyl-glutaryl-coenzyme A reductase, statins can participate in the phenomenon. Antibodies adhered to the myocytes activate the complement cascade, with membrane attack complex formation and myofiber necrosis [98,102]. Infiltrating lymphocytes overexpress programmed cell death protein 1 and become anergic and apoptotic (Knauss) [103], so the infiltrate is mainly composed of necrosis-induced macrophages. Autoantibodies also induce the overexpression of atrogenes, promoting early-onset muscle atrophy [100]. (**b**) Marked muscle weakness and atrophy are the hallmarks of IMNM. Although pathogenic mechanisms do not satisfactorily explain the disease topography, unlike the other idiopathic inflammatory myopathies, IMNM has a predilection for the adductor and flexor compartments of the thighs, with sparing of the semitendinosus [116]. Overt heart failure is unusual, but subclinical myocardial involvement can occur [111,112,113,114]. **C’**: aggregated complement fractions; **C1q**: complement 1q component; **MAC**: membrane attack complex; **MP**: macrophage; **NMAb**: necrotizing myopathy autoantibodies.

## Data Availability

All the gathered data are available in this manuscript.

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
