# Peer review of "Idiopathic Inflammatory Myopathies: Recent Evidence Linking Pathogenesis and Clinical Features"

_ijms, 2025, doi:10.3390/ijms26073302_

Round 1

Reviewer 1 Report

Comments and Suggestions for Authors

Overall this is a fairly comprehensive review of the idiopathic inflammatory myopathy field, covering many key aspects. The manuscript is well written and accessible to both clinicians and scientists, who may be on the periphery of the IIM field.

I have a few suggestions in terms of content, which could be further elaborated on.

My main critique is that the authors predominantly cite review articles, rather than the original research papers throughout the manuscript. This creates a significant issue for the reader when looking to read more broadly and refer back to key papers. This needs addressing throughout the article.

There has been some more recent work on the role of mitochondria in the pathogenesis of IIMs, it may be worth including that - an updated PubMed search would provide those key papers. Secondly, the non-immune aspect of weakness/dysfunction is a little underdeveloped - again, a modest addition to the manuscript could address that aspect. There are further updates around type I and type II IFN's which may be relevant, there is crossover here with the mitochondria work, particularly in sIBM.

Author Response

- Overall this is a fairly comprehensive review of the idiopathic inflammatory myopathy field, covering many key aspects. The manuscript is well written and accessible to both clinicians and scientists, who may be on the periphery of the IIM field. I have a few suggestions in terms of content, which could be further elaborated on.

We truly appreciate the time spent reviewing our manuscript, the positive feedback, and the valuable suggestions. Please find our responses below.

- My main critique is that the authors predominantly cite review articles, rather than the original research papers throughout the manuscript. This creates a significant issue for the reader when looking to read more broadly and refer back to key papers. This needs addressing throughout the article.

We understand your concern. Our objective in citing only the latest review articles was to maintain the reference list updated. However, we are aware that this approach creates authorship issues and may preclude proper retrieval of the primary articles. Therefore, we included new references throughout the manuscript to address this specific issue. 

There has been some more recent work on the role of mitochondria in the pathogenesis of IIMs, it may be worth including that - an updated PubMed search would provide those key papers. Secondly, the non-immune aspect of weakness/dysfunction is a little underdeveloped - again, a modest addition to the manuscript could address that aspect. There are further updates around type I and type II IFN's which may be relevant, there is crossover here with the mitochondria work, particularly in sIBM.

Thank you for your valuable input; we learned a lot from it. We did our best to include more information regarding mitochondrial dysfunction in DM, adding two new paragraphs as follows:

“The role of mitochondrial dysfunction in the pathogenesis of DM has been a growing area of interest(17). This assumption is supported by: 1) mitochondrial DNA (mtDNA) gene variants(18) and mtDNA depletion(19) reported in DM patients, par-ticularly in the perifascicular region, where vasculopathy is most prominent; 2) the presence of surrogate markers of oxidative phosphorylation impairment, such as re-duced ATP production and proton efflux from muscle fibers(20); 3) decreased expres-sion of genes related to electron transport chain complexes in DM(21), along with an increased proportion of cytochrome C oxidase (COX)-negative fibers, the latter corre-lating with reduced aerobic capacity(21); 4) in animal models of DM, a correlation between the IFNγ cytokine profile—one of the inflammatory signatures in DM(22)—and reduced mitochondrial gene expression, as well as diminished expres-sion of genes related to oxidative phosphorylation following IFNγ exposure(23); and 5) the presence of autoantibodies directed against mitochondria in IIM(24)(25)(26), alt-hough not always associated with mitochondrial dysfunction(27).

Despite the compelling evidence linking mitochondrial dysfunction to DM, it re-mains arguable whether the immune attack is the primary driver of mitochondrial homeostasis disruption, leading to reactive oxygen species-mediated injury, dysfunc-tion, and even organelle death. When mtDNA and intracellular components are pre-sented to pattern-recognition receptors, type I IFN may exacerbate inflammation(17) and potentially contribute to the development of autoantibodies.”

Regarding sIBM, a whole paragraph has already been dedicated to mitochondria. We have included additional references and a highlight regarding the type I IFN response:

“Of note, as in DM, several danger-associated molecular patterns—including their highly methylated DNA, cardiolipin, and N-formyl methionine peptides—can initiate or enhance the type I IFN response, perpetuating the inflammatory cycle(17).”

Reviewer 2 Report

Comments and Suggestions for Authors

This review aims to present the recent insights into different myositis subtypes pathogenesis and linking it with clinical manifestations. The review is clear relevant and updated citations. The conclusion are drawn coherent and supported by listed citations. Figures and tables are clear and understandable. I have minor comments:

1- In figure 2 lower panel, Please switch figure 2D and 2C

2- In the conclusion, The authors emphasized using of humoral autoimmunity targeting, however they didn't list any current treatment of any type of IIM except the first one-Dematomyisitis. I think that author should mention Briefly the current treatment used in each IIM. 

3- Page 10 line 337, please remove "1.5 clinical correlations as it is not separate point from previous IMNM paragraphs, then renumber headlines. 

Author Response

- This review aims to present the recent insights into different myositis subtypes pathogenesis and linking it with clinical manifestations. The review is clear relevant and updated citations. The conclusion are drawn coherent and supported by listed citations. Figures and tables are clear and understandable. I have minor comments:

We are truly thankful for your recognition and support. Please find our corrections below.

1- In figure 2 lower panel, Please switch figure 2D and 2C

Ok. It has been switched.

2- In the conclusion, The authors emphasized using of humoral autoimmunity targeting, however they didn't list any current treatment of any type of IIM except the first one-Dematomyisitis. I think that author should mention Briefly the current treatment used in each IIM.

Sure, we agree. We added a brief section in each part addressing the treatment of the respective condition. Of note, we avoided suggesting immunosuppression for sIBM, as its benefit is only marginal.

3- Page 10 line 337, please remove "1.5 clinical correlations as it is not separate point from previous IMNM paragraphs, then renumber headlines.

Ok. Done.